# Exponential expressivity in deep neural networks through transient chaos

**Ben Poole[1], Subhaneil Lahiri[1], Maithra Raghu[2], Jascha Sohl-Dickstein[2], Surya Ganguli[1]**
[1]Stanford University, [2]Google Brain
{benpoole,sulahiri,sganguli}@stanford.edu, {maithra,jaschasd}@google.com

## Abstract

We combine Riemannian geometry with the mean field theory of high dimensional chaos to study the nature of signal propagation in generic, deep neural networks with random weights. Our results reveal an order-to-chaos expressivity phase transition, with networks in the chaotic phase computing nonlinear functions whose global curvature grows *exponentially* with depth but not width. We prove this generic class of deep random functions cannot be efficiently computed by any shallow network, going beyond prior work restricted to the analysis of single functions. Moreover, we formalize and quantitatively demonstrate the long conjectured idea that deep networks can disentangle highly curved manifolds in input space into flat manifolds in hidden space. Our theoretical analysis of the expressive power of deep networks broadly applies to arbitrary nonlinearities, and provides a quantitative underpinning for previously abstract notions about the geometry of deep functions.

## 1 Introduction

Deep feedforward neural networks have achieved remarkable performance across many domains [1–6]. A key factor thought to underlie their success is their high *expressivity*. This informal notion has manifested itself primarily in two forms of intuition. The first is that deep networks can compactly express highly complex functions over input space in a way that shallow networks with one hidden layer and the same number of neurons cannot. The second piece of intuition, which has captured the imagination of machine learning [7] and neuroscience [8] alike, is that deep neural networks can *disentangle* highly curved manifolds in input space into flattened manifolds in hidden space. These intuitions, while attractive, have been difficult to formalize mathematically and thus test rigorously.

For the first intuition, seminal works have exhibited examples of particular functions that can be computed with a polynomial number of neurons (in the input dimension) in a deep network but require an exponential number of neurons in a shallow network [9–13]. This raises a central open question: are such functions merely rare curiosities, or is any function computed by a generic deep network not efficiently computable by a shallow network? The theoretical techniques employed in prior work both limited the applicability of theory to specific nonlinearities and dictated the particular measure of deep functional complexity involved. For example, [9] focused on ReLU nonlinearities and number of linear regions as a complexity measure, while [10] focused on sum-product networks and the number of monomials as complexity measure, and [14] focused on Pfaffian nonlinearities and topological measures of complexity, like the sum of Betti numbers of a decision boundary (however, see [15] for an interesting analysis of a general class of compositional functions). The limits of prior theoretical techniques raise another central question: is there a unifying theoretical framework for deep neural expressivity that is simultaneously applicable to arbitrary nonlinearities, generic networks, and a natural, general measure of functional complexity?

Code to reproduce all results available at: `https://github.com/ganguli-lab/deepchaos`

Here we attack both central problems of deep neural expressivity by combining Riemannian geometry [16] and dynamical mean field theory [17]. This novel combination of tools enables us to show that for very broad classes of nonlinearities, even random deep neural networks can construct hidden internal representations whose global extrinsic curvature grows exponentially with depth but not width. Our geometric framework enables us to quantitatively define a notion of disentangling and verify this notion in deep random networks. Furthermore, our methods yield insights into the emergent, deterministic nature of signal propagation through large random feedforward networks, revealing the existence of an order to chaos transition as a function of the statistics of weights and biases. We find that the transient, finite depth evolution in the chaotic regime underlies the origins of exponential expressivity in deep random networks. In a companion paper [18], we study several related measures of expressivity in deep random neural networks with piecewise linear activations.

## 2  A mean field theory of deep nonlinear signal propagation

Consider a deep feedforward network with $D$ layers of weights $\mathbf{W}^1, \ldots, \mathbf{W}^D$ and $D + 1$ layers of neural activity vectors $\mathbf{x}^0, \ldots, \mathbf{x}^D$, with $N_l$ neurons in each layer $l$, so that $\mathbf{x}^l \in \mathbb{R}^{N_l}$ and $\mathbf{W}^l$ is an $N_l \times N_{l-1}$ weight matrix. The feedforward dynamics elicited by an input $\mathbf{x}^0$ is given by

$$\mathbf{x}^l = \phi(\mathbf{h}^l) \qquad \mathbf{h}^l = \mathbf{W}^l \mathbf{x}^{l-1} + \mathbf{b}^l \quad \text{for } l = 1, \ldots, D, \tag{1}$$

where $\mathbf{b}^l$ is a vector of biases, $\mathbf{h}^l$ is the pattern of inputs to neurons at layer $l$, and $\phi$ is a single neuron scalar nonlinearity that acts component-wise to transform inputs $\mathbf{h}^l$ to activities $\mathbf{x}^l$. We wish to understand the nature of typical functions computable by such networks, as a consequence of their depth. We therefore study ensembles of random networks in which each of the synaptic weights $\mathbf{W}^l_{ij}$ are drawn i.i.d. from a zero mean Gaussian with variance $\sigma_w^2 / N_{l-1}$, while the biases are drawn i.i.d. from a zero mean Gaussian with variance $\sigma_b^2$. This weight scaling ensures that the input contribution to each individual neuron at layer $l$ from activities in layer $l-1$ remains $O(1)$, independent of the layer width $N_{l-1}$. This ensemble constitutes a maximum entropy distribution over deep neural networks, subject to constraints on the means and variances of weights and biases. This ensemble induces no further structure in the resulting set of deep functions, so its analysis provides an opportunity to understand the specific contribution of depth alone to the nature of typical functions computed by deep networks.

In the limit of large layer widths, $N_l \gg 1$, certain aspects of signal propagation through deep random neural networks take on an essentially deterministic character. This emergent determinism in large random neural networks enables us to understand how the Riemannian geometry of simple manifolds in the input layer $\mathbf{x}^0$ is typically modified as the manifold propagates into the deep layers. For example, consider the simplest case of a single input vector $\mathbf{x}^0$. As it propagates through the network, its length in downstream layers will change. We track this changing length by computing the normalized squared length of the input vector at each layer:

$$q^l = \frac{1}{N_l} \sum_{i=1}^{N_l} (\mathbf{h}^l_i)^2. \tag{2}$$

This length is the second moment of the empirical distribution of inputs $\mathbf{h}^l_i$ across all $N_l$ neurons in layer $l$. For large $N_l$, this empirical distribution converges to a zero mean Gaussian since each $\mathbf{h}^l_i = \sum_j \mathbf{W}^l_{ij} \phi(\mathbf{h}^{l-1}_j) + \mathbf{b}^l_i$ is a weighted sum of a large number of uncorrelated random variables - i.e. the weights $\mathbf{W}^l_{ij}$ and biases $\mathbf{b}^l_i$, which are independent of the activity in previous layers. By propagating this Gaussian distribution across one layer, we obtain an iterative map for $q^l$ in (2):

$$q^l = \mathcal{V}(q^{l-1} \mid \sigma_w, \sigma_b) \equiv \sigma_w^2 \int \mathcal{D}z \, \phi \left( \sqrt{q^{l-1}} z \right)^2 + \sigma_b^2, \quad \text{for} \quad l = 2, \ldots, D, \tag{3}$$

where $\mathcal{D}z = \frac{dz}{\sqrt{2\pi}} e^{-\frac{z^2}{2}}$ is the standard Gaussian measure, and the initial condition is $q^1 = \sigma_w^2 q^0 + \sigma_b^2$, where $q^0 = \frac{1}{N_0} \mathbf{x}^0 \cdot \mathbf{x}^0$ is the length in the initial activity layer. See Supplementary Material (SM) for a derivation of (3). Intuitively, the integral over $z$ in (3) replaces an average over the empirical distribution of $\mathbf{h}^l_i$ across neurons $i$ in layer $l$ at large layer width $N_l$.

The function $\mathcal{V}$ in (3) is an iterative variance, or length, map that predicts how the length of an input in (2) changes as it propagates through the network. This length map is plotted in Fig. 1A for the special

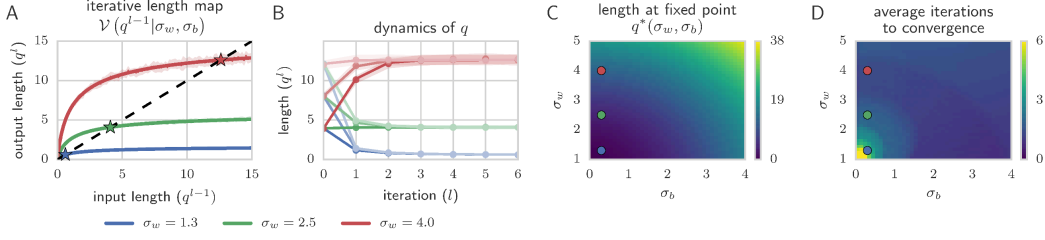

Figure 1: Dynamics of the squared length $q^l$ for a sigmoidal network ($\phi(h) = \tanh(h)$) with 1000 hidden units. (A) The iterative length map in (3) for 3 different $\sigma_w$ at $\sigma_b = 0.3$. Theoretical predictions (solid lines) match well with individual network simulations (dots). Stars reflect fixed points $q^*$ of the map. (B) The iterative dynamics of the length map yields rapid convergence of $q^l$ to its fixed point $q^*$, independent of initial condition (lines=theory; dots=simulation). (C) $q^*$ as a function of $\sigma_w$ and $\sigma_b$. (D) Number of iterations required to achieve $\leq 1\%$ fractional deviation off the fixed point. The $(\sigma_b, \sigma_w)$ pairs in (A,B) are marked with color matched circles in (C,D).

case of a sigmoidal nonlinearity, $\phi(h) = \tanh(h)$. For monotonic nonlinearities, this length map is a monotonically increasing, concave function whose intersections with the unity line determine its fixed points $q^*(\sigma_w, \sigma_b)$. For $\sigma_b = 0$ and $\sigma_w < 1$, the only intersection is at $q^* = 0$. In this bias-free, small weight regime, the network shrinks all inputs to the origin. For $\sigma_w > 1$ and $\sigma_b = 0$, the $q^* = 0$ fixed point becomes unstable and the length map acquires a second nonzero fixed point, which is stable. In this bias-free, large weight regime, the network expands small inputs and contracts large inputs. Also, for any nonzero bias $\sigma_b$, the length map has a single stable non-zero fixed point. In such a regime, even with small weights, the injected biases at each layer prevent signals from decaying to 0. The dynamics of the length map leads to rapid convergence of length to its fixed point with depth (Fig. 1B,D), often within only 4 layers. The fixed points $q^*(\sigma_w, \sigma_b)$ are shown in Fig. 1C.

## 3    Transient chaos in deep networks

Now consider the layer-wise propagation of two inputs $\mathbf{x}^{0,1}$ and $\mathbf{x}^{0,2}$. The geometry of these two inputs as they propagate through the network is captured by the 2 by 2 matrix of inner products:

$$q_{ab}^l = \frac{1}{N_l} \sum_{i=1}^{N_l} \mathbf{h}_i^l(\mathbf{x}^{0,a}) \, \mathbf{h}_i^l(\mathbf{x}^{0,b}) \quad a,b \in \{1,2\}. \tag{4}$$

The dynamics of the two diagonal terms are each theoretically predicted by the length map in (3). We derive (see SM) a correlation map $\mathcal{C}$ that predicts the layer-wise dynamics of $q_{12}^l$:

$$q_{12}^l = \mathcal{C}(c_{12}^{l-1}, q_{11}^{l-1}, q_{22}^{l-1} \mid \sigma_w, \sigma_b) \equiv \sigma_w^2 \int \mathcal{D}z_1 \, \mathcal{D}z_2 \, \phi(u_1)\, \phi(u_2) + \sigma_b^2, \tag{5}$$

$$u_1 = \sqrt{q_{11}^{l-1}}\, z_1, \qquad u_2 = \sqrt{q_{22}^{l-1}}\left[ c_{12}^{l-1} z_1 + \sqrt{1 - (c_{12}^{l-1})^2}\, z_2 \right],$$

where $c_{12}^l = q_{12}^l (q_{11}^l q_{22}^l)^{-1/2}$ is the correlation coefficient. Here $z_1$ and $z_2$ are independent standard Gaussian variables, while $u_1$ and $u_2$ are correlated Gaussian variables with covariance matrix $\langle u_a u_b \rangle = q_{ab}^{l-1}$. Together, (3) and (5) constitute a theoretical prediction for the typical evolution of the geometry of 2 points in (4) in a fixed large network.

Analysis of these equations reveals an interesting order to chaos transition in the $\sigma_w$ and $\sigma_b$ plane. In particular, what happens to two nearby points as they propagate through the layers? Their relation to each other can be tracked by the correlation coefficient $c_{12}^l$ between the two points, which approaches a fixed point $c^*(\sigma_w, \sigma_b)$ at large depth. Since the length of each point rapidly converges to $q^*(\sigma_w, \sigma_b)$, as shown in Fig. 1BD, we can compute $c^*$ by simply setting $q_{11}^l = q_{22}^l = q^*(\sigma_w, \sigma_b)$ in (5) and dividing by $q^*$ to obtain an iterative correlation coefficient map, or $\mathcal{C}$-map, for $c_{12}^l$:

$$c_{12}^l = \frac{1}{q^*} \mathcal{C}(c_{12}^{l-1}, q^*, q^* \mid \sigma_w, \sigma_b). \tag{6}$$

This $\mathcal{C}$-map is shown in Fig. 2A. It always has a fixed point at $c^* = 1$ as can be checked by direct calculation. However, the stability of this fixed point depends on the slope of the map at 1, which is

$$\chi_1 \equiv \left.\frac{\partial c_{12}^l}{\partial c_{12}^{l-1}}\right|_{c=1} = \sigma_w^2 \int \mathcal{D}z \left[\phi'\left(\sqrt{q^*}z\right)\right]^2. \tag{7}$$

See SM for a derivation of (7). If the slope $\chi_1$ is less than 1, then the $\mathcal{C}$-map is above the unity line, the fixed point at 1 under the $\mathcal{C}$-map in (6) is stable, and nearby points become more similar over time.

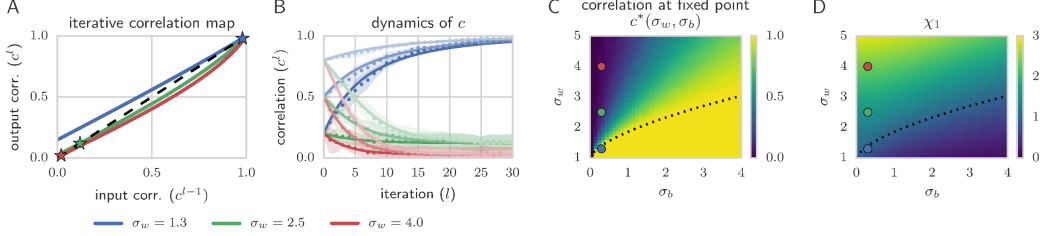

Figure 2: Dynamics of correlations, $c_{12}^l$, in a sigmoidal network with $\phi(h) = \tanh(h)$. (A) The $\mathcal{C}$-map in (6) for the same $\sigma_w$ and $\sigma_b = 0.3$ as in Fig. 1A. (B) The $\mathcal{C}$-map dynamics, derived from both theory, through (6) (solid lines) and numerical simulations of (1) with $N_l = 1000$ (dots) (C) Fixed points $c^*$ of the $\mathcal{C}$-map. (D) The slope of the $\mathcal{C}$-map at 1, $\chi_1$, partitions the space (black dotted line at $\chi_1 = 1$) into chaotic ($\chi_1 > 1$, $c^* < 1$) and ordered ($\chi_1 < 1$, $c^* = 1$) regions.

Conversely, if $\chi_1 > 1$ then this fixed point is unstable, and nearby points separate as they propagate through the layers. Thus we can intuitively understand $\chi_1$ as a multiplicative stretch factor. This intuition can be made precise by considering the Jacobian $\mathbf{J}_{ij}^l = \mathbf{W}_{ij}^l \phi'(\mathbf{h}_j^{l-1})$ at a point $\mathbf{h}^{l-1}$ with length $q^*$. $\mathbf{J}^l$ is a linear approximation of the network map from layer $l-1$ to $l$ in the vicinity of $\mathbf{h}^{l-1}$. Therefore a small random perturbation $\mathbf{h}^{l-1} + \mathbf{u}$ will map to $\mathbf{h}^l + \mathbf{J}\mathbf{u}$. The growth of the perturbation, $||\mathbf{J}\mathbf{u}||_2^2/||\mathbf{u}||_2^2$ becomes $\chi_1(q^*)$ after averaging over the random perturbation $\mathbf{u}$, weight matrix $\mathbf{W}^l$, and Gaussian distribution of $\mathbf{h}_i^{l-1}$ across $i$. Thus $\chi_1$ directly reflects the typical multiplicative growth or shrinkage of a random perturbation across one layer.

The dynamics of the iterative $\mathcal{C}$-map and its agreement with network simulations is shown in Fig. 2B. The correlation dynamics are much slower than the length dynamics because the $\mathcal{C}$-map is closer to the unity line (Fig. 2A) than the length map (Fig. 1A). Thus correlations typically take about 20 layers to approach the fixed point, while lengths need only 4. The fixed point $c^*$ and slope $\chi_1$ of the $\mathcal{C}$-map are shown in Fig. 2CD. For any fixed, finite $\sigma_b$, as $\sigma_w$ increases three qualitative regions occur. For small $\sigma_w$, $c^* = 1$ is the only fixed point, and it is stable because $\chi_1 < 1$. In this strong bias regime, any two input points converge to each other as they propagate through the network. As $\sigma_w$ increases, $\chi_1$ increases and crosses 1, destabilizing the $c^* = 1$ fixed point. In this intermediate regime, a new stable fixed point $c^*$ appears, which decreases as $\sigma_w$ increases. Here an equal footing competition between weights and nonlinearities (which de-correlate inputs) and the biases (which correlate them), leads to a finite $c^*$. At larger $\sigma_w$, the strong weights overwhelm the biases and maximally de-correlate inputs to make them orthogonal, leading to a stable fixed point at $c^* = 0$.

Thus the equation $\chi_1(\sigma_w, \sigma_b) = 1$ yields a phase transition boundary in the $(\sigma_w, \sigma_b)$ plane, separating it into a chaotic (or ordered) phase, in which nearby points separate (or converge). In dynamical systems theory, the logarithm of $\chi_1$ is related to the well known Lyapunov exponent which is positive (or negative) for chaotic (or ordered) dynamics. However, in a feedforward network, the dynamics is truncated at a finite depth $D$, and hence the dynamics are a form of transient chaos.

## 4  The propagation of manifold geometry through deep networks

Now consider a 1 dimensional manifold $\mathbf{x}^0(\theta)$ in input space, where $\theta$ is an intrinsic scalar coordinate on the manifold. This manifold propagates to a new manifold $\mathbf{h}^l(\theta) = \mathbf{h}^l(\mathbf{x}^0(\theta))$ in the vector space of inputs to layer $l$. The typical geometry of the manifold in the $l$'th layer is summarized by $q^l(\theta_1, \theta_2)$, which for any $\theta_1$ and $\theta_2$ is defined by (4) with the choice $\mathbf{x}^{0,a} = \mathbf{x}^0(\theta_1)$ and $\mathbf{x}^{0,b} =$

$\mathbf{x}^0(\theta_2)$. The theory for the propagation of pairs of points applies to all pairs of points on the manifold, so intuitively, we expect that in the chaotic phase of a sigmoidal network, the manifold should in some sense de-correlate, and become more complex, while in the ordered phase the manifold should contract around a central point. This theoretical prediction of equations (3) and (5) is quantitatively confirmed in simulations in Fig. 3, when the input is a simple manifold, the circle, $\mathbf{h}^1(\theta) = \sqrt{N_1 q}\left[\mathbf{u}^0 \cos(\theta) + \mathbf{u}^1 \sin(\theta)\right]$, where $\mathbf{u}^0$ and $\mathbf{u}^1$ form an orthonormal basis for a 2 dimensional subspace of $\mathbb{R}^{N_1}$ in which the circle lives. The scaling is chosen so that each neuron has input activity $O(1)$. Also, for simplicity, we choose the fixed point radius $q = q^*$ in Fig. 3.

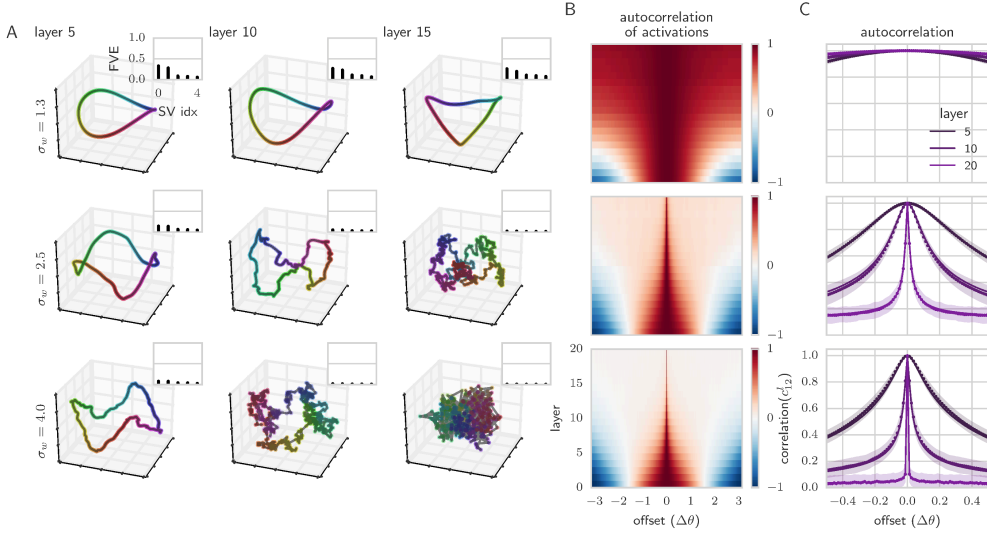

Figure 3: Propagating a circle through three random sigmoidal networks with varying $\sigma_w$ and fixed $\sigma_b = 0.3$. (A) Projection of hidden inputs of simulated networks at layer 5 and 10 onto their first three principal components. Insets show the fraction of variance explained by the first 5 singular values. For large weights (bottom), the distribution of singular values gets flatter and the projected curve is more tangled. (B) The autocorrelation, $c_{12}^l(\Delta\theta) = \int d\theta\, q^l(\theta, \theta + \Delta\theta)/q^*$, of hidden inputs as a function of layer for simulated networks. (C) The theoretical predictions from (6) (solid lines) compared to the average (dots) and standard deviation across $\theta$ (shaded) in a simulated network.

To quantitatively understand the layer-wise growth of complexity of this manifold, it is useful to turn to concepts in Riemannian geometry [16]. First, at each point $\theta$, the manifold $\mathbf{h}(\theta)$ (we temporarily suppress the layer index $l$) has a tangent, or velocity vector $\mathbf{v}(\theta) = \partial_\theta \mathbf{h}(\theta)$. Intuitively, curvature is related to how quickly this tangent vector rotates in the ambient space $\mathbb{R}^N$ as one moves along the manifold, or in essence the acceleration vector $\mathbf{a}(\theta) = \partial_\theta \mathbf{v}(\theta)$. Now at each point $\theta$, when both are nonzero, $\mathbf{v}(\theta)$ and $\mathbf{a}(\theta)$ span a 2 dimensional subspace of $\mathbb{R}^N$. Within this subspace, there is a unique circle of radius $R(\theta)$ that has the same position, velocity and acceleration vector as the curve $\mathbf{h}(\theta)$ at $\theta$. This circle is known as the osculating circle (Fig. 4A), and the extrinsic curvature $\kappa(\theta)$ of the curve is defined as $\kappa(\theta) = 1/R(\theta)$. Thus, intuitively, small radii of curvature $R(\theta)$ imply high extrinsic curvature $\kappa(\theta)$. The extrinsic curvature of a curve depends only on its image in $\mathbb{R}^N$ and is invariant with respect to the particular parameterization $\theta \to \mathbf{h}(\theta)$. For any parameterization, an explicit expression for $\kappa(\theta)$ is given by $\kappa(\theta) = (\mathbf{v} \cdot \mathbf{v})^{-3/2}\sqrt{(\mathbf{v} \cdot \mathbf{v})(\mathbf{a} \cdot \mathbf{a}) - (\mathbf{v} \cdot \mathbf{a})^2}$ [16]. Note that under a unit speed parameterization of the curve, so that $\mathbf{v}(\theta) \cdot \mathbf{v}(\theta) = 1$, we have $\mathbf{v}(\theta) \cdot \mathbf{a}(\theta) = 0$, and $\kappa(\theta)$ is simply the norm of the acceleration vector.

Another measure of the curve's complexity is the length $\mathcal{L}^E$ of its image in the ambient Euclidean space. The Euclidean metric in $\mathbb{R}^N$ induces a metric $g^E(\theta) = \mathbf{v}(\theta) \cdot \mathbf{v}(\theta)$ on the curve, so that the distance $d\mathcal{L}^E$ moved in $\mathbb{R}^N$ as one moves from $\theta$ to $\theta + d\theta$ on the curve is $d\mathcal{L}^E = \sqrt{g^E(\theta)}d\theta$. The total curve length is $\mathcal{L}^E = \int \sqrt{g^E(\theta)}d\theta$. However, even straight line segments can have a large Euclidean length. Another interesting measure of length that takes into account curvature, is the length of the image of the curve under the Gauss map. For a $K$ dimensional manifold $\mathcal{M}$ embedded in

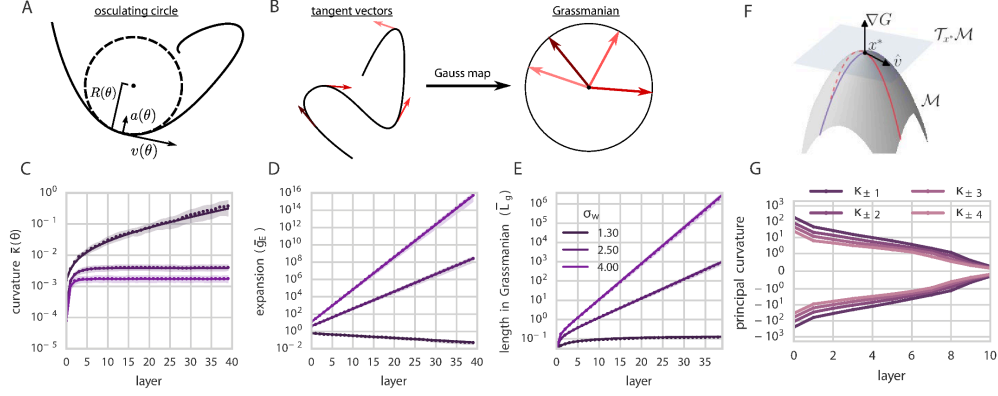

Figure 4: Propagation of extrinsic curvature and length in a network with 1000 hidden units. (A) An osculating circle. (B) A curve with unit tangent vectors at 4 points in ambient space, and the image of these points under the Gauss map. (C-E) Propagation of curvature metrics based on both theory derived from iterative maps in (3), (6) and (8) (solid lines) and simulations using (1) (dots). (F) Schematic of the normal vector, tangent plane, and principal curvatures for a 2D manifold embedded in $\mathbb{R}^3$. (G) average principal curvatures for the largest and smallest 4 principal curvatures ($\kappa_{\pm 1}, \ldots, \kappa_{\pm 4}$) across locations $\theta$ within one network. The principal curvatures all grow exponentially as we backpropagate to the input layer. Panels F,G are discussed in Sec. 5.

$\mathbb{R}^N$, the Gauss map (Fig. 4B) maps a point $\theta \in \mathcal{M}$ to its $K$ dimensional tangent plane $\mathcal{T}_\theta \mathcal{M} \in \mathfrak{G}_{K,N}$, where $\mathfrak{G}_{K,N}$ is the Grassmannian manifold of all $K$ dimensional subspaces in $\mathbb{R}^N$. In the special case of $K = 1$, $\mathfrak{G}_{K,N}$ is the sphere $\mathbb{S}^{N-1}$ with antipodal points identified, since a 1-dimensional subspace can be identified with a unit vector, modulo sign. The Gauss map takes a point $\theta$ on the curve and maps it to the unit velocity vector $\hat{\mathbf{v}}(\theta) = \mathbf{v}(\theta)/\sqrt{\mathbf{v}(\theta) \cdot \mathbf{v}(\theta)}$. In particular, the natural metric on $\mathbb{S}^{N-1}$ induces a Gauss metric on the curve, given by $g^G(\theta) = (\partial_\theta \hat{\mathbf{v}}(\theta)) \cdot (\partial_\theta \hat{\mathbf{v}}(\theta))$, which measures how quickly the unit tangent vector $\hat{\mathbf{v}}(\theta)$ changes as $\theta$ changes. Thus the distance $d\mathcal{L}^G$ moved in the Grassmannian $\mathfrak{G}_{K,N}$ as one moves from $\theta$ to $\theta + d\theta$ on the curve is $d\mathcal{L}^G = \sqrt{g^G(\theta)}d\theta$, and the length of the curve under the Gauss map is $\mathcal{L}^G = \int \sqrt{g^G(\theta)}d\theta$. Furthermore, the Gauss metric is related to the extrinsic curvature and the Euclidean metric via the relation $g^G(\theta) = \kappa(\theta)^2 g^E(\theta)$ [16].

To illustrate these concepts, it is useful to compute all of them for the circle $\mathbf{h}^1(\theta)$ defined above: $g^E(\theta) = Nq$, $\mathcal{L}^E = 2\pi\sqrt{Nq}$, $\kappa(\theta) = 1/\sqrt{Nq}$, $g^G(\theta) = 1$, and $\mathcal{L}^G = 2\pi$. As expected, $\kappa(\theta)$ is the inverse of the radius of curvature, which is $\sqrt{Nq}$. Now consider how these quantities change if the circle is scaled up so that $\mathbf{h}(\theta) \to \chi\mathbf{h}(\theta)$. The length $\mathcal{L}^E$ and radius scale up by $\chi$, but the curvature $\kappa$ scales down as $\chi^{-1}$, and so $\mathcal{L}^G$ does not change. Thus linear expansion increases length and decreases curvature, thereby maintaining constant Grassmannian length $\mathcal{L}^G$.

We now show that nonlinear propagation of this same circle through a deep network can behave very differently from linear expansion: in the chaotic regime, length can increase without any decrease in extrinsic curvature! To remove the scaling with $N$ in the above quantities, we will work with the renormalized quantities $\bar{\kappa} = \sqrt{N}\kappa$, $\bar{g}^E = \frac{1}{N}g^E$, and $\bar{\mathcal{L}}^E = \frac{1}{\sqrt{N}}\mathcal{L}^E$. Thus, $1/(\bar{\kappa})^2$ can be thought of as a radius of curvature squared per neuron of the osculating circle, while $(\bar{\mathcal{L}}^E)^2$ is the squared Euclidean length of the curve per neuron. For the circle, these quantities are $q$ and $2\pi q$ respectively. For simplicity, in the inputs to the first layer of neurons, we begin with a circle $\mathbf{h}^1(\theta)$ with squared radius per neuron $q^1 = q^*$, so this radius is already at the fixed point of the length map in (3). In the SM, we derive an iterative formula for the extrinsic curvature and Euclidean metric of this manifold as it propagates through the layers of a deep network:

$$\bar{g}^{E,l} = \chi_1 \, \bar{g}^{E,l-1} \qquad (\bar{\kappa}^l)^2 = 3\frac{\chi_2}{\chi_1^2} + \frac{1}{\chi_1}(\bar{\kappa}^{l-1})^2, \qquad \bar{g}^{E,1} = q^*, \qquad (\bar{\kappa}^1)^2 = 1/q^*. \qquad (8)$$

where $\chi_1$ is the stretch factor defined in (7) and $\chi_2$ is defined analogously as

$$\chi_2 = \sigma_w^2 \int \mathcal{D}z \left[\phi''\left(\sqrt{q^*}z\right)\right]^2. \qquad (9)$$

$\chi_2$ is closely related to the second derivative of the $\mathcal{C}$-map in (6) at $c_{12}^{l-1} = 1$; this second derivative is $\chi_2 q^*$. See SM for a derivation of the evolution equations for extrinsic geometry in (8).

Intriguingly for a sigmoidal neural network, these evolution equations behave very differently in the chaotic ($\chi_1 > 1$) versus ordered ($\chi_1 < 1$) phase. In the chaotic phase, the Euclidean metric $\bar{g}^E$ grows *exponentially* with depth due to multiplicative stretching through $\chi_1$. This stretching does multiplicatively attenuate any curvature in layer $l - 1$ by a factor $1/\chi_1$ (see the update equation for $\bar{\kappa}^l$ in (8)), but new curvature is added in due to a nonzero $\chi_2$, which originates from the curvature of the single neuron nonlinearity in (9). Thus, unlike in linear expansion, extrinsic curvature is not lost, but maintained, and ultimately approaches a fixed point $\bar{\kappa}^*$. This implies that the global curvature measure $\bar{\mathcal{L}}^G$ grows exponentially with depth. These highly nontrivial predictions of the metric and curvature evolution equations in (8) are quantitatively confirmed in simulations in Figure 4C-E.

Intuitively, this exponential growth of global curvature $\bar{\mathcal{L}}^G$ in the chaotic phase implies that the curve explores many different tangent directions in hidden representation space. This further implies that the coordinate functions of the embedding $\mathbf{h}_i^l(\theta)$ become highly complex curved basis functions on the input manifold coordinate $\theta$, allowing a deep network to compute exponentially complex functions over simple low dimensional manifolds (Figure 5A-C, details in SM).

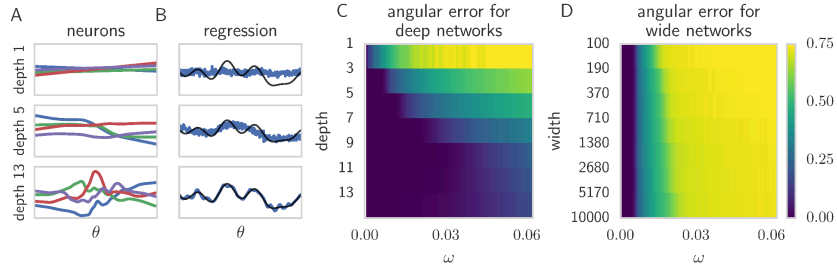

Figure 5: Deep networks in the chaotic regime are more expressive than shallow networks. (A) Activity of four different neurons in the output layer as a function of the input, $\theta$ for three networks of different depth (width $N_l = 1,000$). (B) Linear regression of the output activity onto a random function (black) shows closer predictions (blue) with deeper networks (bottom) than shallow networks (top). (C) Decomposing the prediction error by frequency shows shallow networks cannot capture high frequency content in random functions but deep networks can (yellow=high error). (D) Increasing the width of a one hidden layer network up to $10,000$ does not decrease error at high frequencies.

## 5   Shallow networks cannot achieve exponential expressivity

Consider a shallow network with 1 hidden layer $\mathbf{x}^1$, one input layer $\mathbf{x}^0$, with $\mathbf{x}^1 = \phi(\mathbf{W}^1\mathbf{x}^0) + \mathbf{b}^1$, and a linear readout layer. How complex can the hidden representation be as a function of its width $N_1$, relative to the results above for depth? We prove a general upper bound on $\mathcal{L}^E$ (see SM):

**Theorem 1.** *Suppose $\phi(h)$ is monotonically non-decreasing with bounded dynamic range $R$, i.e. $\max_h \phi(h) - \min_h \phi(h) = R$. Further suppose that $\mathbf{x}^0(\theta)$ is a curve in input space such that no 1D projection of $\partial_\theta \mathbf{x}(\theta)$ changes sign more than $s$ times over the range of $\theta$. Then for any choice of $\mathbf{W}^1$ and $\mathbf{b}^1$ the Euclidean length of $\mathbf{x}^1(\theta)$, satisfies $\mathcal{L}^E \le N_1(1 + s)R$.*

For the circle input, $s = 1$ and for the $\tanh$ nonlinearity, $R = 2$, so in this special case, the normalized length $\bar{\mathcal{L}}^E \le 2\sqrt{N_1}$. In contrast, for deep networks in the chaotic regime $\bar{\mathcal{L}}^E$ grows exponentially with depth in $\mathbf{h}$ space, and so consequently also in $\mathbf{x}$ space. Therefore the length of curves typically expand exponentially in depth even for random deep networks, but can only expand as the square root of width no matter what shallow network is chosen. Moreover, as we have seen above, it is the exponential growth of $\bar{\mathcal{L}}^{\mathcal{E}}$ that fundamentally drives the exponential growth of $\bar{\mathcal{L}}^G$ with depth. Indeed shallow random networks exhibit minimal growth in expressivity even at large widths (Figure 5D).

## 6   Classification boundaries acquire exponential local curvature with depth

We have focused so far on how simple manifolds in input space can acquire both exponential Euclidean and Grassmannian length with depth, thereby exponentially de-correlating and filling up

hidden representation space. Another natural question is how the complexity of a decision boundary grows as it is backpropagated to the input layer. Consider a linear classifier $y = \text{sgn}(\boldsymbol{\beta} \cdot \mathbf{x}^D - \beta_0)$ acting on the final layer. In this layer, the $N-1$ dimensional decision boundary is the hyperplane $\boldsymbol{\beta} \cdot \mathbf{x}^D - \beta_0 = 0$. However, in the input layer $\mathbf{x}^0$, the decision boundary is a curved $N-1$ dimensional manifold $\mathcal{M}$ that arises as the solution set of the nonlinear equation $G(\mathbf{x}^0) \equiv \boldsymbol{\beta} \cdot \mathbf{x}^D(\mathbf{x}^0) - \beta_0 = 0$, where $\mathbf{x}^D(\mathbf{x}^0)$ is the nonlinear feedforward map from input to output.

At any point $\mathbf{x}^*$ on the decision boundary in layer $l$, the gradient $\vec{\nabla} G$ is perpendicular to the $N-1$ dimensional tangent plane $\mathcal{T}_{x^*}\mathcal{M}$ (see Fig. 4F). The normal vector $\vec{\nabla} G$, along with any unit tangent vector $\hat{\mathbf{v}} \in \mathcal{T}_{x^*}\mathcal{M}$, spans a 2 dimensional subspace whose intersection with $\mathcal{M}$ yields a geodesic curve in $\mathcal{M}$ passing through $x^*$ with velocity vector $\hat{\mathbf{v}}$. This geodesic will have extrinsic curvature $\kappa(\mathbf{x}^*, \hat{\mathbf{v}})$. Maximizing this curvature over $\hat{\mathbf{v}}$ yields the first principal curvature $\kappa_1(\mathbf{x}^*)$. A sequence of successive maximizations of $\kappa(\mathbf{x}^*, \hat{\mathbf{v}})$, while constraining $\hat{\mathbf{v}}$ to be perpendicular to all previous solutions, yields the sequence of principal curvatures $\kappa_1(\mathbf{x}^*) \geq \kappa_2(\mathbf{x}^*) \geq \cdots \geq \kappa_{N-1}(\mathbf{x}^*)$. These principal curvatures arise as the eigenvalues of a normalized Hessian operator projected onto the tangent plane $\mathcal{T}_{x^*}\mathcal{M}$: $\mathcal{H} = ||\vec{\nabla} G||_2^{-1} \mathbf{P} \frac{\partial^2 G}{\partial \mathbf{x} \partial \mathbf{x}^T} \mathbf{P}$, where $\mathbf{P} = \mathbf{I} - \widehat{\nabla} G \widehat{\nabla} G^T$ is the projection operator onto $\mathcal{T}_{x^*}\mathcal{M}$ and $\widehat{\nabla} G$ is the unit normal vector [16]. Intuitively, near $\mathbf{x}^*$, the decision boundary $\mathcal{M}$ can be approximated as a paraboloid with a quadratic form $\mathcal{H}$ whose $N-1$ eigenvalues are the principal curvatures $\kappa_1, \ldots, \kappa_{N-1}$ (Fig. 4F).

We compute these curvatures numerically as a function of depth in Fig. 4G (see SM for details). We find, remarkably, that a subset of principal curvatures grow *exponentially* with depth. Here the principal curvatures are signed, with positive (negative) curvature indicating that the associated geodesic curves towards (away from) the normal vector $\vec{\nabla} G$. Thus the decision boundary can become exponentially curved with depth, enabling highly complex classifications. Moreover, this exponentially curved boundary is disentangled and mapped to a flat boundary in the output layer.

## 7 Discussion

Fundamentally, neural networks compute nonlinear maps between high dimensional spaces, for example from $\mathbb{R}^{N_1} \to \mathbb{R}^{N_D}$, and it is unclear what the most appropriate mathematics is for understanding such daunting spaces of maps. Previous works have attacked this problem by restricting the nature of the nonlinearity involved (e.g. piecewise linear, sum-product, or Pfaffian) and thereby restricting the space of maps to those amenable to special theoretical analysis methods (combinatorics, polynomial relations, or topological invariants). We have begun a preliminary exploration of the expressivity of such deep functions based on Riemannian geometry and dynamical mean field theory. We demonstrate that networks in a chaotic phase compactly exhibit functions that exponentially grow the global curvature of simple one dimensional manifolds from input to output and the local curvature of simple co-dimension one manifolds from output to input. The former captures the notion that deep neural networks can efficiently compute highly expressive functions in ways that shallow networks cannot, while the latter quantifies and demonstrates the power of deep neural networks to disentangle curved input manifolds, an attractive idea that has eluded formal quantification.

Moreover, our analysis of a maximum entropy distribution over deep networks constitutes an important null model of deep signal propagation that can be used to assess and understand different behavior in trained networks. For example, the metrics we have adapted from Riemannian geometry, combined with an understanding of their behavior in random networks, may provide a basis for understanding what is special about trained networks. Furthermore, while we have focused on the notion of input-output chaos, the duality between inputs and synaptic weights imply a form of weight chaos, in which deep neural networks rapidly traverse function space as weights change (see SM). Indeed, just as autocorrelation lengths between outputs as a function of inputs shrink exponentially with depth, so too will autocorrelations between outputs as a function of weights. Finally, while our length and correlation maps can be applied directly to piecewise linear nonlinearities (e.g. ReLUs), deep piecewise linear functions have 0 local curvature. To characterize how such functions twist across input space, our methods can compute tangent vector auto-correlations instead of curvature.

But more generally, to understand functions, we often look to their graphs. The graph of a map from $\mathbb{R}^{N_1} \to \mathbb{R}^{N_D}$ is an $\mathbb{R}^{N_1}$ dimensional submanifold of $\mathbb{R}^{N_1+N_D}$, and therefore has both high dimension and co-dimension. We speculate that many of the secrets of deep learning may be uncovered by studying the geometry of this graph as a Riemannian manifold, and understanding how it changes with both depth and learning.

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
