[Supplementary Material]

# Supplementary material for Exponential expressivity in deep neural networks through transient chaos

**Ben Poole**[1], **Subhaneil Lahiri**[1], **Maithra Raghu**[2], **Jascha Sohl-Dickstein**[2], **Surya Ganguli**[1]
[1]Stanford University, [2]Google Brain
{benpoole,sulahiri,sganguli}@stanford.edu, {maithra,jaschasd}@google.com

Below is a series of appendices giving derivations of results in the main paper, followed by details of results along with more visualizations.

## 1 Derivation of a transient dynamical mean field theory for deep networks

We study a deep feedforward network with $D$ layers of weights $\mathbf{W}^1, \ldots, \mathbf{W}^D$ and $D+1$ layers of neural activity vectors $\mathbf{x}^0, \ldots, \mathbf{x}^D$, with $N_l$ neurons in each layer $l$, so that $\mathbf{x}^l \in \mathbb{R}^{N_l}$ and $\mathbf{W}^l$ is an $N_l \times N_{l-1}$ weight matrix. The feedforward dynamics elicited by an input $\mathbf{x}^0$ is given by

$$\mathbf{x}^l = \phi(\mathbf{h}^l) \qquad \mathbf{h}^l = \mathbf{W}^l \mathbf{x}^{l-1} + \mathbf{b}^l \quad \text{for } l = 1, \ldots, D, \tag{1}$$

where $\mathbf{b}^l$ is a vector of biases, $\mathbf{h}^l$ is the pattern of inputs to neurons at layer $l$, and $\phi$ is a single neuron scalar nonlinearity that acts component-wise to transform inputs $\mathbf{h}^l$ to activities $\mathbf{x}^l$. The synaptic weights $\mathbf{W}^l_{ij}$ are drawn i.i.d. from a zero mean Gaussian with variance $\sigma_w^2/N_{l-1}$, while the biases are drawn i.i.d. from a zero mean Gaussian with variance $\sigma_b^2$. This weight scaling ensures that the input contribution to each individual neuron at layer $l$ from activities in layer $l-1$ remains $O(1)$, independent of the layer width $N_{l-1}$.

### 1.1 Derivation of the length map

As a single input point $\mathbf{x}^0$ propagates through the network, it's length in downstream layers can either grow or shrink. To track the propagation of this length, we track the normalized squared length of the input vector at each layer,

$$q^l = \frac{1}{N_l} \sum_{i=1}^{N_l} (\mathbf{h}_i^l)^2. \tag{2}$$

This length is the second moment of the empirical distribution of inputs $\mathbf{h}_i^l$ across all $N_l$ neurons in layer $l$ for a *fixed* set of weights. This empirical distribution is expected to be Gaussian for large $N_l$, since each individual $\mathbf{h}_i^l = \mathbf{w}^{l,i} \cdot \phi(\mathbf{h}^{l-1}) + \mathbf{b}_i^l$ is Gaussian distributed, as a sum of a large number of independent random variables, and each $\mathbf{h}_i^l$ is independent of $\mathbf{h}_j^l$ for $i \neq j$ because the synaptic weights vectors and biases into each neuron are chosen independently.

While the mean of this Gaussian is 0, its variance can be computed by considering the variance of the input to a single neuron:

$$q^l = \langle (\mathbf{h}_i^l)^2 \rangle = \left\langle \left[ \mathbf{w}^{l,i} \cdot \phi(\mathbf{h}^{l-1}) \right]^2 \right\rangle + \langle (\mathbf{b}_i^l)^2 \rangle = \sigma_w^2 \frac{1}{N_{l-1}} \sum_{i=1}^{N_{l-1}} \phi(\mathbf{h}_i^{l-1})^2 + \sigma_b^2, \tag{3}$$

where $\langle \cdot \rangle$ denotes an average over the distribution of weights and biases into neuron $i$ at layer $l$. Here we have used the identity $\langle \mathbf{w}_j^{l,i} \mathbf{w}_k^{l,i} \rangle = \delta_{jk} \, \sigma_w^2/N_{l-1}$. Now the empirical distribution of inputs across layer $l-1$ is also Gaussian, with mean zero and variance $q^{l-1}$. Therefore we can replace the average

over neurons in layer $l-1$ in (3) with an integral over a Gaussian random variable, obtaining

$$q^l = \mathcal{V}(q^{l-1} \,|\, \sigma_w, \sigma_b) \equiv \sigma_w^2 \int \mathcal{D}z \, \phi \left( \sqrt{q^{l-1}} z \right)^2 + \sigma_b^2, \quad \text{for} \quad l = 2, \ldots, D, \tag{4}$$

where $\mathcal{D}z = \frac{dz}{\sqrt{2\pi}} e^{-\frac{z^2}{2}}$ is the standard Gaussian measure, and the initial condition for the variance map is $q^1 = \sigma_w^2 q_0 + \sigma_b^2$, where $q^0 = \frac{1}{N_0} \mathbf{x}^0 \cdot \mathbf{x}^0$ is the length in the initial activity layer. The function $\mathcal{V}$ in (4) is an iterative variance map that predicts how the length of an input in (2) changes as it propagates through the network. Its derivation relies on the well-known self-averaging assumption in the statistical physics of disordered systems, which, in our context, means that the empirical distribution of inputs *across* neurons for a *fixed* network converges for large width, to the distribution of inputs to a *single* neuron *across* random networks.

## 1.2 Derivation of a correlation map for the propagation of two points

Now consider the layer-wise propagation of two inputs $\mathbf{x}^{0,1}$ and $\mathbf{x}^{0,2}$. The geometry of these two inputs as they propagate through the layers is captured by the 2 by 2 matrix of inner products

$$q_{ab}^l = \frac{1}{N_l} \sum_{i=1}^{N_l} \mathbf{h}_i^l(\mathbf{x}^{0,a}) \, \mathbf{h}_i^l(\mathbf{x}^{0,b}) \quad a, b \in \{1, 2\}. \tag{5}$$

The *joint* empirical distribution of $\mathbf{h}_i^l(\mathbf{x}^{0,a})$ and $\mathbf{h}_i^l(\mathbf{x}^{0,a})$ across $i$ at large $N_l$ will converge to a 2 dimensional Gaussian distribution with covariance $q_{ab}^l$. Propagating this joint distribution forward one layer using ideas similar to the derivation above for 1 input yields

$$q_{12}^l = \mathcal{C}(c_{12}^{l-1}, q_{11}^{l-1}, q_{22}^{l-1} \,|\, \sigma_w, \sigma_b) \equiv \sigma_w^2 \int \mathcal{D}z_1 \, \mathcal{D}z_2 \, \phi(u_1) \, \phi(u_2) + \sigma_b^2, \tag{6}$$

$$u_1 = \sqrt{q_{11}^{l-1}} z_1, \qquad u_2 = \sqrt{q_{22}^{l-1}} \left[ c_{12}^{l-1} z_1 + \sqrt{1 - (c_{12}^{l-1})^2} z_2 \right],$$

where $c_{12}^l = \frac{q_{12}^l}{\sqrt{q_{11}^l} \sqrt{q_{12}^l}}$ is the correlation coefficient (CC). Here $z_1$ and $z_2$ are independent standard Gaussian variables, while $u_1$ and $u_2$ are correlated Gaussian variables with covariance matrix $\langle u_a u_b \rangle = q_{ab}^{l-1}$. The integration over $z_1$ and $z_2$ can be thought of as the large $N_l$ limit of sums over $\mathbf{h}_i^l(\mathbf{x}^{0,a})$ and $\mathbf{h}_i^l(\mathbf{x}^{0,b})$.

When both input points are at their fixed point length, $q^*$, the dynamics of their correlation coefficient can be obtained by simply setting $q_{11}^l = q_{22}^l = q^*(\sigma_w, \sigma_b)$ in (6) and dividing by $q^*$ to obtain a recursion relation for $c_{12}^l$:

$$c_{12}^l = \frac{1}{q^*} \mathcal{C}(c_{12}^{l-1}, q^*, q^* \,|\, \sigma_w, \sigma_b) \tag{7}$$

Direct calculation reveals that $c_{12}^l(1) = 1$ as expected. Of particular interest is the slope $\chi_1$ of this map at 1. A direct, if tedious calculation shows that

$$\frac{\partial c_{12}^l}{\partial c_{12}^{l-1}} = \sigma_w^2 \int \mathcal{D}z_1 \, \mathcal{D}z_2 \, \phi'(u_1) \, \phi'(u_2). \tag{8}$$

To obtain this result, one has to apply the chain rule and product rule from calculus, as well as employ the identity

$$\int \mathcal{D}z F(z) z = \int \mathcal{D}z F'(z), \tag{9}$$

which can be obtained via integration by parts. Evaluating the derivative at 1 yields

$$\chi_1 \equiv \frac{\partial c_{12}^l}{\partial c_{12}^{l-1}} \bigg|_{c=1} = \sigma_w^2 \int \mathcal{D}z \left[ \phi'\left( \sqrt{q^*} z \right) \right]^2. \tag{10}$$

# 2 Derivation of evolution equations for Riemannian curvature

Here we derive recursion relations for Riemannian curvature quantitites.

## 2.1 Curvature and length in terms of inner products

Consider a translation invariant manifold, or 1D curve $\mathbf{h}(\theta) \in \mathbb{R}^N$ that is on some constant radius sphere so that

$$q(\theta_1, \theta_2) = Q(\theta_1 - \theta_2) = \mathbf{h}(\theta_1) \cdot \mathbf{h}(\theta_2), \tag{11}$$

with $Q(0) = Nq^*$. At large $N$, the inner-product structure of translation invariant manifolds remains approximately translation invariant as it propagates through the network. Therefore, at large $N$, we can express inner products of derivatives of $\mathbf{h}$ in terms of derivatives of $Q$. For example, the Euclidean metric $g^E$ is given by

$$g^E(\theta) = \partial_\theta \mathbf{h}(\theta) \cdot \partial_\theta \mathbf{h}(\theta) = -\ddot{Q}(0). \tag{12}$$

Here, each dot is a short hand notation for derivative w.r.t. $\theta$. Also, the extrinsic curvature

$$\kappa(\theta) = \sqrt{\frac{(\mathbf{v} \cdot \mathbf{v})(\mathbf{a} \cdot \mathbf{a}) - (\mathbf{v} \cdot \mathbf{a})^2}{(\mathbf{v} \cdot \mathbf{v})^3}}, \tag{13}$$

where $\mathbf{v}(\theta) = \partial_\theta \mathbf{h}(\theta)$ and $\mathbf{a}(\theta) = \partial_\theta^2 \mathbf{h}(\theta)$, simplifies to

$$\kappa(\theta) = \frac{\ddddot{Q}(0)}{\ddot{Q}(0)^2}. \tag{14}$$

Now if the translation invariant manifold lives on a sphere of radius $Nq^*$ where $q^*$ is the fixed point radius of the length map, then its radius does not change as it propagates through the system. Then we can also express $g^E$ and $\kappa$ in terms of the correlation coefficient function $c(\theta) = Q(\theta)/q^*$ (up to a factor of $N$). Thus to understand the propagation of local quantities like Euclidean length and curvature, we need to understand the propagation of derivatives of $c(\theta)$ at $\theta = 0$ under the $\mathcal{C}$-map in (7). Note that $c(\theta)$ is symmetric and achieves a maximum value of 1 at $\theta = 0$. Thus the function $H^1(\theta) = 1 - c(\theta)$ is symmetric with a minimum at $\theta = 0$. We consider the propagation of $H^1$ though the $\mathcal{C}$-map. But first we consider the propagation of derivatives under function composition in general.

## 2.2 Behavior of first and second derivatives under function composition

Assume $H^1(\Delta t)$ is an even function and $H^1(0) = 0$, so that its Taylor expansion can be written as $H^1(\Delta t) = \frac{1}{2}\ddot{H}^1(0)\Delta t^2 + \frac{1}{4}\ddddot{H}^1(0)\Delta t^4 + \dots$. We are interested in determining how the second and fourth derivatives of $H$ propagate under composition with another function $G$, so that $H^2 = G(H^1(\Delta t))$. We assume $G(0) = 0$. We can use the chain rule and the product rule to derive:

$$\ddot{H}^2(0) = \dot{G}(0)\ddot{H}^1(0) \tag{15}$$

$$\ddddot{H}^2(0) = 3\ddot{G}(0)\ddot{H}^1(0)^2 + \dot{G}(0)\ddddot{H}^1(0). \tag{16}$$

## 2.3 Evolution equations for curvature and length

We now apply the above iterations with $H^1(\theta) = 1 - c(\theta)$ and $G(c) = 1 - \frac{1}{q^*}\mathcal{C}(1 - c, q^*, q^* \mid \sigma_w, \sigma_b)$. Clearly, $G(0) = 0$ the symmetric $H^1$ obeys $H^1(0) = 0$, satisfying the above iterations of second and fourth derivatives. Taking into account these derivative recursions, using the expressions for $\kappa$ and $g^E$ in terms of derivatives of $c(\theta)$ at 0, and carefully accounting for factors of $q^*$ and $N$, we obtain the final evolution equations that have been successfully tested against experiments:

$$\bar{g}^{E,l} = \chi_1 \bar{g}^{E,l-1} \tag{17}$$

$$(\bar{\kappa}^l)^2 = 3\frac{\chi_2}{\chi_1^2} + \frac{1}{\chi_1}(\bar{\kappa}^{l-1})^2, \tag{18}$$

where $\chi_1$ is the stretch factor defined in (10) and $\chi_2$ is defined analogously as

$$\chi_2 = \sigma_w^2 \int \mathcal{D}z \left[\phi''\left(\sqrt{q^*}z\right)\right]^2. \tag{19}$$

$\chi_2$ is closely related to the second derivative of the correlation coefficient map in (7) at $c_{12}^{l-1} = 1$. Indeed this second derivative is $\chi_2 q^*$.

# 3 Upper bounds on the complexity of shallow neural representations

Consider a shallow network with 1 hidden layer $\mathbf{x}^1$ and one input layer $\mathbf{x}^0$, so that $\mathbf{x}^1 = \phi(\mathbf{W}^1\mathbf{x}^0) + \mathbf{b}$. The network can compute functions through a linear readout of the hidden layer $\mathbf{x}^1$. We are interested in how complex these neural representations can get, with one layer of synaptic weights and nonlinearities, as a function the number of hidden units $N_1$. In particular, we are interested in how the length and curvature of an input manifold $\mathbf{x}^0(\theta)$ changes as it propagates to become $\mathbf{x}^1(\theta)$ in the hidden layer. We would like to upper bound the maximal achievable length and curvature over all possible choices of $\mathbf{W}^1$ and $\mathbf{b}$.

## 3.1 Upper bound on Euclidean length

Here, we derive such an upper bound on the Euclidean length for a very general class of nonlinearities $\phi(h)$. We simply assume that (1) $\phi(h)$ is monotonically non-decreasing (so that $\phi'(h) \geq 0 \forall h$) and (2) has with bounded dynamic range R, i.e. $\max_h \phi(h) - \min_h \phi(h) = R$. The Euclidean length in hidden space is

$$\mathcal{L}^E \quad = \quad \int d\theta \sqrt{\sum_{i=1}^{N_1} (\partial_\theta \mathbf{x}_i^1(\theta))^2} \quad \leq \quad \sum_{i=1}^{N_1} \int d\theta \left| \partial_\theta \mathbf{x}_i^1(\theta) \right|, \tag{20}$$

where the inequality follows from the triangle inequality. Now suppose that for any $i$, $\partial_\theta \mathbf{x}_i^1(\theta)$ never changes sign across $\theta$. Furthermore, assume that $\theta$ ranges from 0 to $\Theta$. Then

$$\int_0^\Theta d\theta \left| \partial_\theta \mathbf{x}_i^1(\theta) \right| = \mathbf{x}_i^1(\Theta) - \mathbf{x}_i^1(0) \leq \left( \max_h \phi(h) - \min_h \phi(h) \right) = R. \tag{21}$$

More generally, let $r_1$ denote the maximal number of times that any one neuron has a change in sign of the derivative $\partial_\theta \mathbf{x}_i^1(\theta)$ across $\theta$. Then applying the above argument to each segment of constant sign yields

$$\int_0^\Theta d\theta \left| \partial_\theta \mathbf{x}_i^1(\theta) \right| \leq (1 + r_1)R. \tag{22}$$

Now how many times can $\partial_\theta \mathbf{x}_i^1(\theta)$ change sign? Since $\partial_\theta \mathbf{x}_i^1(\theta) = \phi'(\mathbf{h}_i) \partial_\theta \mathbf{h}_i$, where $\partial_\theta \mathbf{h}_i(\theta) = [\mathbf{W}^l \partial_\theta \mathbf{x}^0(\theta)]_i$, and $\phi(\mathbf{h}_i)$ is monotonically increasing, the number of times $\partial_\theta \mathbf{x}_i^1(\theta)$ changes sign equals the number of times the input $\partial_\theta \mathbf{h}_i(\theta)$ changes sign. In turn, suppose $s_0$ is the maximal number of times any one dimensional projection of the derivative vector $\partial_\theta \mathbf{x}^0(\theta)$ changes sign across $\theta$. Then the number of times the sign of $\partial_\theta \mathbf{h}_i(\theta)$ changes for any $i$ cannot exceed $s_0$ because $\mathbf{h}_i$ is a linear projection of $\mathbf{x}^0$. Together this implies $r_1 \leq s_0$. We have thus proven:

$$\mathcal{L}^E \leq N_1(1 + s_0)R. \tag{23}$$

# 4 Simulation details

All neural network simulations were implemented in Keras and Theano. For all simulations (except Figure 5C), we used inputs and hidden layers with a width of 1,000 and tanh activations. We found that our results were mostly insensitive to width, but using larger widths decreased the fluctuations in the averaged quantities. Simulation error bars are all standard deviations, with the variance computed across the different inputs, $h^1(\theta)$. If not mentioned, the weights in the network are initialized in the chaotic regime with $\sigma_b = 0.3$, $\sigma_w = 4.0$.

Computing $\kappa(\theta)$ requires the computation of the velocity and acceleration vectors, corresponding to the first and second derivatives of the neural network $h^l(\theta)$ with respect to $\theta$. As $\theta$ is always one-dimensional, we can greatly speed up these computations by using forward-mode auto-differentiation, evaluating the Jacobian and Hessian in a feedforward manner. We implemented this using the R-op in Theano.

## 4.1 Details on Figure 4G: backpropagating curvature

To identify the curvature of the decision boundary, we first had to identify points that lied along the decision boundary. We randomly initialized data points and then optimized $G(x^D(x^l))^2$ with respect

to the input $x$ using Adam. This yields a set of inputs $x^l$ where we compute the Jacobian and Hessian of $G(x^D(x^l))$ to evaluate principal curvatures.

## 4.2 Details on Figure 5C-D: evaluating expressivity

To evaluate the set of functions reachable by a network, we first parameterized function space using a Fourier basis up to a particular maximum frequency, $\omega_{\max}$ on a sampled set of one dimensional inputs parameterized by $\theta$. We then took the output activations of each neural network and linearly regressed the output activations onto each Fourier basis. For each basis, we computed the angle between the predicted basis vector and the true basis vector. These are the quantities that appear in Figure 5C-D. Given any function with bounded frequency, we can represent it in this Fourier basis, and decompose the error in the prediction of the function into the error in prediction of each Fourier component. Thus error in the predicting the Fourier basis is a reasonable proxy for error in prediction of functions with bounded frequency.

# 5 Additional visualization of hidden actions

Figure 1: Two-dimensional t-SNE embeddings of hidden activations of a network in the chaotic regime. These can be compared to the PCA visualizations in Fig. 3A of the main paper. Our results reveal that the decreasing radii of curvature of both the t-SNE plots and PCA plots with depth is an illusion, or artifact of attempting to fit a fundamentally high dimensional curve into a low dimensional space. Instead, our curvature evolution equations reveal the geometric nature of curve without resorting to dimensionality reduction. As the depth increases, the radius of curvature stays *constant* while the length grows *exponentially*. As a result, the curve makes a number of turns that is exponentially large in depth, where each turn occurs at a depth independent acceleration. In this manner, the curve fills up the high dimensional hidden representation space. The t-SNE plot faithfully reflects the number of turns, but not their radius of curvature.

# 6 A view from the function space perspective

We have shown above that for a fixed set of weights and biases in the chaotic regime, the internal representation $\mathbf{h}^l(\mathbf{x}^0)$ at large depth $l$, rapidly de-correlates from itself as the input $\mathbf{x}^0$ changes (see e.g. Fig. 3B in the main paper). Here we ask a dual question: for a fixed input manifold, how does a deep network move in a function space over this manifold as the weights in a single layer change? Consider for example, a random one parameter family of deep networks parameterized by $\Delta \in [-1, 1]$. In this family, we assume that the bias vectors $\mathbf{b}^l$ in each layer are chosen as i.i.d. random Gaussian vectors with zero mean and variance $\sigma_b^2$, independent of $\Delta$. Moreover, we assume the weight matrix $\mathbf{W}^l$ has elements that are drawn i.i.d. from zero mean Gaussians with variance $\sigma_w^2$, independent of $\Delta$ for all layers except $l = 2$. The only dependence on $\Delta$ in this family of networks originates in the weights in layer $l = 2$, chosen as

$$\mathbf{W}^l(\Delta) = \sqrt{1 - |\Delta|}\,\mathbf{W} + \sqrt{|\Delta|}\,\mathbf{dW}. \tag{24}$$

Here both a base matrix $\mathbf{W}$ and a perturbation matrix $\mathbf{dW}$ have matrix elements that are zero mean i.i.d. Gaussians with variance $\sigma_w^2$. Each matrix element of $\mathbf{W}^2(\Delta)$ thus also has variance $\sigma_w^2$ just like all the other layers. In turn, this family of networks induces a family of functions $\mathbf{h}^D(\mathbf{h}^1, \Delta)$. For simplicity, we restrict these functions to a simple input manifold, the circle,

$$\mathbf{h}^1(\theta) = \sqrt{N_1 q^*}\left[\mathbf{u}^0 \cos(\theta) + \mathbf{u}^1 \sin(\theta)\right], \tag{25}$$

as considered previously. This circle is at the fixed point radius $q^*(\sigma_w, \sigma_b)$, and the family of networks induces a family of functions from the circle to the hidden representation space in layer $l$, namely $\mathbb{R}^{N_l}$. We denote these functions by $\mathbf{h}^l(\theta, \Delta)$. How similar are these functions as $\Delta$ changes? This can be quantified through the correlation in function space

$$Q^l(\Delta_1, \Delta_2) \equiv \int \frac{d\theta}{2\pi} \frac{1}{N_D} \sum_{i=1}^{N_D} \mathbf{h}_i^l(\theta, \Delta_1)\mathbf{h}_i^l(\theta, \Delta_2), \tag{26}$$

and the associated correlation coefficient,

$$C^l(\Delta) = \frac{Q^l(0, \Delta)}{\sqrt{Q^l(0, 0)Q^l(\Delta, \Delta)}}. \tag{27}$$

Because of our restriction to an input circle at the fixed point radius, $Q^l(0, 0) = Q^l(\Delta, \Delta) = q^*$ for all $l$ and $\Delta$ in the large width limit. By using logic similar to the derivation of (6), we can derive a recursion relation for the function space correlation $Q^l(0, \Delta)$:

$$Q^l(0, \Delta) = \sigma_w^2 \int \mathcal{D}z_1\,\mathcal{D}z_2\,\phi(u_1)\,\phi(u_2) + \sigma_b^2, \quad l = 3, \ldots, D \tag{28}$$

$$Q^l(0, \Delta) = \sqrt{1 - |\Delta|}\sigma_w^2 \int \mathcal{D}z_1\,\mathcal{D}z_2\,\phi(u_1)\,\phi(u_2) + \sigma_b^2, \quad l = 2,$$

$$u_1 = \sqrt{q^*}z_1, \qquad u_2 = \sqrt{q^*}\left[C^{l-1}(\Delta)z_1 + \sqrt{1 - (C^{l-1}(\Delta))^2}z_2\right],$$

where $C^l(\Delta) = Q^l(0, \Delta)/q^*$. The initial condition for this recursion is $C^1(\Delta) = 1$, since the family of functions in the first layer of inputs is independent of $\Delta$. Now, the difference in weights at a nonzero $\Delta$ reduces the function space correlation to $C^2(\Delta) < 1$. At this point, the representation in $\mathbf{h}^2$ is different for the two networks at parameter values $0$ and $\Delta$. Moreover, in the chaotic regime, this difference will amplify due to the similarity between the function space evolution equation in (28) and the evolution equation for the similarity of two points in (6). In essence, just as two points in the input exponentially separate as they propagate through a single network in the chaotic regime, a pair of different functions separate when computed in the final layer. Thus a small perturbation in the weights into layer 2 can yield a very large change in the space of functions from the input manifold to layer $D$. Moreover, as $\Delta$ varies from -1 to 1, the function $\mathbf{h}^D(\theta, \Delta)$ roughly undergoes a random walk in function space whose autocorrelation length decreases exponentially with depth $D$. This weight chaos, or a sensitive dependence of the function computed by a deep network with respect to weight changes far from the final layer, is another manifestation of deep neural expressivity. Our companion paper **?** further explores the expressivity of deep random networks in function space and also finds an exponential growth in expressivity with depth.