[Reviews · NeurIPS 2016]

Reviewer 1

Summary

This paper studies differences between deep and shallow neural networks. It shows that certain neural networks with random weights compute functions whose curvature grows exponentially with depth but not with width.

Qualitative Assessment

* The paper focusses on curvature and normally distributed weights. This is an interesting direction for studying the differences between the functions computed by deep and shallow neural networks. * I think that the description of previous works in the Introduction and Abstract could be more accurate. For instance, `have exhibited examples of particular functions' should actually read `have exhibited classes of functions'. In particular, [5] discusses functions computed by deep networks which require exponentially larger shallow networks and which are stable under perturbation of parameters, meaning that these functions are generic within the considered deep networks. * The paper could comment on the role of differentiability assumptions for the analysis and how it would apply to activation functions that are not everywhere differentiable. * The first section is missing a title. * line 220. there is a typo `expand expand'

Confidence in this Review

2-Confident (read it all; understood it all reasonably well)


Reviewer 2

Summary

The main results in this paper: 1) Deep neural networks are expressive. Specifically, an L-deep network will typically (with the right scaling of the weights) map any 1D curve in its input to a curve which has non-vanishing curvature and euclidian length exponentially increasing in L. 2) Shallow neural networks are not expressive. Specifically, for any given 1D curve in the input, a bound is derived on the elucidation length of the curve at the hidden layer. Thus, deep networks are more expressive then shallow networks.

Qualitative Assessment

This is a very interesting work. However, I have a few major concerns: 1) I believe Theorem 1 is wrong, as can be seen from the counterexample at the bottom of this review. As can be observed from this counterexample, the main problem in the proof is the inaccurate sentence on lines 110-112 in the supplementary material. I'll wait to author's feedback before deciding if this a fatal flaw. 2) I believe the statement that h_i^l are uncorrelated random variables (line 69 in paper + line 22 in supplementary material) is wrong for l > 1 (i.e., second hidden layer, or deeper). In this case, h_i^l are all composed of different linear sums of the same random vector x^{l-1}, and are therefore dependent. I agree this dependency should vanish in the limit of large N and random W, but this should be clearly stated and quantified (to prevent potential issues when applying similar calculations for trained networks). 3) The authors should explain the novelty of their results in comparison to those in the recent ICML paper “Convolutional Rectifier Networks as Generalized Tensor Decompositions” by Cohen et al. Minor comments: 4) Section 1 is very similar to section G in the supplementary material of “Exact solutions to the nonlinear dynamics of learning in deep linear neural networks” by Saxe et al. I feel this similarity should be stated and discussed. For example, Saxe et al. suggested that it is good to initialize near the transition to Chaos. It would be nice if the authors can clarify if this is related to the results in the current paper. 5) It would be nice if the authors could clarify how one should to use eq. (9) for non-differentiable activation functions (e.g., the popular ReLU). 6) Most of the paper is written very clearly, except Section 5, which feels a bit rushed. I think it should be improved or removed. 7) There are a few typos. For example, in the caption of fig. 3: “5 and 10” -> “5,10 and 15” “ 6 “ -> “ (6)” (in two places). *** Counterexample *** x^0(t) = ( -2t , 2t+sin(t) ) which is a curve over some range 0 < t < 2*pi*M, where M is an positive integer (and we replaced \theta from the paper with t). Taking the derivative: d x^0(t) / dt = ( -2 , 2+cos(t) ) we see that s=0. Choosing N1=1,W=(1,1), b=0 and activation function \phi(h) = min(1,max(-1,h)) (so R=2) we get d x^1(t) / dt = d h^1(t) / dt = cos(t) Therefore, the total Euclidian length is L^E=int _ 0 ^ {2*pi*M} |cos(t)| dt = 4*M , which is larger than the bound in Theorem 1: N1*(1+s)*R = 1*1*2 = 2. %% After Rebuttal %% 1) Correction of Theorem 1 looks good. I hoped this would be possible. 2) Yes, I was wrong here. Sorry for the confusion. 3) Sure, I understand your method is more true for more general activation functions. Still it would be very helpful to the reader if the authors place their results within the right context. For example, if you restrict yourself to the setting (ReLU+MaxPooling) in Cohen&Shashua 2016, can you get different/better results? Thanks for this nice paper!

Confidence in this Review

2-Confident (read it all; understood it all reasonably well)


Reviewer 3

Summary

The authors study the input-output map of a function that is obtained by a feed-forward deep neural network with random iid weights as each level. They focus on the study of how the distance between two close by inputs changes as the input propagates trough the network. They identify an ordered phase where the distance decreases to zero, and a chaotic phase where the distance grows with depth. They argue that finite depth evolution the chaotic phase underlies the origin of exponential expressivity in deep networks.

Qualitative Assessment

My overall impression is rather mixed. The paper addresses relevant questions. The analysis is clear and well understandable. However, I am doubtful about the relevance of the regime the authors focus at - the finite depth evolution in the chaotic regime. I fail to see how the chaotic regime can be relevant in applications. In all applications I can think of we do require that the output of the network does not change much when the input is perturbed (noisy or deformed cat is still a cat). One main conclusion is that these random functions cannot be efficiently computed by shallow networks, but why would we want to compute such chaotic functions in practice?

Confidence in this Review

2-Confident (read it all; understood it all reasonably well)


Reviewer 4

Summary

The paper analysis the transformation of curves through a feedforward layered network architecture with depth. It presents an interesting approach reminiscent of the analysis of random recurrent networks where the evolution of activity over time is investigated, whereas here the interest in on the depth of the network. The authors show an interesting analogy of the phase transition in chaotic systems usually analysed over time rather than depth. The authors derive equations for the propagation of correlations of any two points through the random network and use this to investigate how curvature of manifolds is persevered or not and show that deep network have an exponential expressivity whereas single layer networks do not.

Qualitative Assessment

I really enjoyed reading the paper. It uses an impressive amount of advanced mathematical techniques with ease to show very convincingly the theoretical basis of how deep networks are able to distangle manifolds. While intuitive, the authors present this argument in a stringent yet insightful way in the language of curvature of Riemannian manifolds. It also presents an excellent analysis of fix points and state transitions of the signal propagation through deep networks using dynamical mean-field methods, which might be useful in applications to put constraints on weights and biases. However, it seems that some parts are similar to published papers as one reviewer pointed out.

Confidence in this Review

2-Confident (read it all; understood it all reasonably well)